# Factors Influencing Rooibos Tea Certification and Quality Control for Smallholder Farmers in South Africa

**DOI:** 10.3390/foods11213495

**Published:** 2022-11-03

**Authors:** Aijun Liu, Siphiwe Charmaine Ka Makhaya, Maurice Osewe

**Affiliations:** 1College of Economics and Management, Nanjing Agricultural University, 1 Weigang, Nanjing 210095, China; 2China Center for Food Security Studies, Nanjing Agricultural University, 1 Weigang, Nanjing 210095, China

**Keywords:** certification, quality assurance systems, rooibos tea, off-farm income, farming practice, education level

## Abstract

This study aimed to identify factors that influence the decisions of rooibos farmers in South Africa to implement certification and quality assurance systems. The study was conducted in the Western Cape region of South Africa. A structured questionnaire was distributed to 300 farmers in the form of interviews. In addition, an analysis of previously published data was also used. Results showed that membership in an association, land tenure, rooibos tea farm size, and education level were the main determinants of implementing certifications and quality assurance systems. Membership in the association and land tenure significantly negatively affected the adoption of certification. In contrast, farm size and level of education, translating to knowledge of certification systems, tended to have a significant positive effect on adoption. Continuous education, awareness of the process of certification and quality assurance systems, and the formation of farmers’ support systems are recommended to improve the impact of smallholder rooibos farmers in the industry.

## 1. Introduction

Rooibos (*Aspalathus linearis*), (Burm.f.) R. Dahlgren is a broom-like member of the plant family Fabaceae that grows along the Western Cape’s coastal areas in South Africa [1]. The leaves are used to make an indigenous herbal tea called rooibos which was discovered by Benjamin Ginsberg in 1904 [2]. Commercialization of rooibos tea began in the 1930s [3], and its popularity subsequently increased internationally, with consumption in over 37 countries [4]. In 2010, Germany, the Netherlands, Japan, Poland, the United Kingdom, and the USA made up 86% of rooibos tea exports [1,5]. Rooibos tea is commonly used as a herbal and medicinal tea because of its high antioxidant properties. The tea was also reported to relieve allergies, dermatological problems, asthma, infantile colic, and other gastrointestinal complaints, such as nausea and heartburn [3]. Rooibos tea is also known to improve appetite, reduce tension and improve sleep [6,7]. Despite these unique qualities, the current market share of rooibos tea makes up less than 3% of the total South African tea market [4]. However, globally, rooibos tea contributes to approximately 10% of the herbal tea market, with approximately 6000–7000 tonnes per year. South Africa is the sole exporter of rooibos tea globally. Conventional rooibos tea is the leading exported type compared to the green and organic varieties. Figure 1 illustrates the percentages of rooibos exported to the top destinations.

Tea is one of the world’s most consumed beverages, alongside cocoa and coffee [2]. Hence, the study of rooibos tea qualification and certification is significant as it warrants implementing more robust control measures and providing better-quality tea. There is a potential for sustainable living in rooibos farming for farmers in Clan Williams town, the region with the most smallholder rooibos farmers. However, current rooibos farmers face significant challenges due to shifting requirements and the demand for quality and safety assurances of rooibos by-products [8,9]. Research undertaken by [5] noted that not all farmers had implemented quality assurance practices. This makes it impossible for them to participate in international rooibos markets. Moreover, several types of research have been conducted on quality assurance [3,10,11], yet none explain the factors influencing the implementation of quality assurance systems by rooibos farmers. Therefore, we conducted this research with the hope of understanding these factors.

## 2. Research Methodology

### 2.1. Data Sources and Survey Design

This research was conducted in the Western Cape Province of South Africa. The region is located along the Cederberg Mountains (32.1976° S, 18.8967° E) and was purposely chosen because it is the only region in the world where rooibos is grown. A proportionate random sampling technique was used to sample 300 rooibos farmers. A structured questionnaire was administered, and data were collected, collated, compiled, and verified. The structured questionnaire was developed to collect essential and relevant data from the farmers through interviews. The questionnaire had three sections. The first section contained questions to gather general information such as farm size, tea yield, income distributions, and farming systems used. The second section comprised an in-depth look at the farmers’ knowledge and implementation of quality assurance systems. The third section inquired about the farmers’ demographic characteristics, such as gender, age, household size, land ownership, cooperative membership, education, and off-farm income. The interviews were complemented with information created from a literature review [12]. The factors were organized into statements the respondents could answer by indicating one of the five options on a Likert scale ranging from 5 (strongly agree) to 1 (strongly disagree) in the interview form. Figure 2 illustrates the distribution of rooibos farms around Clanwilliams town; all the farms are situated within the same area.

### 2.2. Empirical Modeling

The logit regression model was used to determine the factors influencing a farmer’s decision to implement a quality assurance system. The logit regression model was selected because it can predict the probability of farmers’ implementing new technologies [13]. The model is based on cumulative logistic probability functions where the dependent variable is measured as dichotomous, implementers or non-implementers. Thus, according to [13,14], the logit model is mathematically expressed as:
$$P_{i}=\frac{eI_{i}}{1-eI_{i}}$$
$$\log\left(\frac{P_{i}}{1-P_{i}}\right)=\beta_{0}+\beta_{0}X_{1}+\mu_{i}$$


And the decision is estimated by:*y* = 1 if *y** > 0, and *y* = 0 if *y** ≤ 0.

That is, 1 = implementers, and 0 = non-implementers (otherwise).

Where Y, measured in dichotomous nature, indicates whether a farmer implemented a certification and quality assurance system or not. This value is the exogenous variable used to indicate certification and quality assurance decisions; *ε* represents the normally distributed error term with zero mean and constant variance, while *β* is the parameter to be estimated. The variables of description are shown in Table 1 below.

Therefore, the final logistic regression equation was modelled as:QA*S_ij_** = *β*_0_ + *β*_1i_GENDER + *β*_2i_AGE + *β*_3i_EDUC + *β*_4i_HHSIZE + *β*_5i_SOF + *β*_6i_TOT + *β*_7i_OFFFARMINC + *β*_8i_Membership + *ε*

## 3. Results

### 3.1. Characteristics of Participants

Table 2 lists the demographic data of the farmers who participated in the survey. The majority of the farmers were males with primary school education. Less than five percent of the farmers attained university degrees. Most of the households consisted of four people, and most of the farmers were members of farmer organizations/societies. The membership in the organization was fairly distributed, with the Wupperthal cooperative having the most farmers. Of these farmers, the majority rented the land or farmed their private land, while only a few were farmed through government initiatives forming part of the cooperatives.

### 3.2. Types of Farming Practices

The rooibos crop is harvested from December to February; therefore, because of this short harvesting period, farmers tend to engage in other farming practices to supplement their livelihoods. Hence, we inquired about the additional type of farming, apart from rooibos, that the farmers engaged in (Table 3). Thirty-one percent of farmers engaged in livestock farming as well as subsistence agriculture and cash crop farming, and the remaining farmers were engaged in other farming practices.

### 3.3. Rooibos Farming Characteristics

Table 4 indicates the size of the rooibos farms, the rooibos yield per hectare of land, and the equivalent income per kilogram of rooibos. The statistics show that cultivated rooibos increase premiums even though the wild type creates market competition. Wild rooibos cost R1.11–R16.31 (R = South African Rand currency), whereas cultivated rooibos cost R3–R45 per kilogram. As a result, cultivated rooibos is the most popular type among farmers and consumers. The cultivated rooibos variety is fast-growing and high-yielding, though less drought and pest resistant than the wild varieties [15].

### 3.4. Rejection of Rooibos and Food Safety Risks Awareness

We also asked the farmers about food safety risks, such as food from unsafe sources, personal hygiene, and market rejections. Most farmers indicated that they had never had their rooibos rejected. The limited awareness of food safety risks reported by a few farmers could be attributed to small-scale farming and lacking access to technology and online search engines.

### 3.5. Rooibos Farmer’s Knowledge of Quality Assurance Systems

Farmers were questioned about their level of knowledge on rooibos quality assurance systems (Figure 3). Rooibos quality assurance includes inspection and testing by the Perishable Products Export Control Board of South Africa and certification by the Directorate of Health and Quality and a Phytosanitary. Surprisingly, more than 30 percent of the farmers were unaware of any quality assurance systems. A few indicated they were not concerned about these systems because consumers purchased the tea without inquiring about it. This is true because individual farmers who cultivate the tea for local consumption sell mainly to loyal local consumers who are not concerned about certification.

### 3.6. Types of Certifications Used by Farmers

Farmers were asked to comment about the types of certifications they currently use to facilitate the export of their products. From our results and those published by [16], we learned that non-governmental organizations assist farmers and members of cooperatives in obtaining certificates that help them export their products. Figure 4 illustrates the distribution of certification types that farmers use to authenticate rooibos tea quality assurance. The majority of the farmers (use certifications from the United States Department of Agriculture (USDA, 26%), followed by the South Africa Bureau of Standards (SABS, 21%), and less than two percent of the farmers did not use any certification.

### 3.7. Reasons for Not Implementing Quality Assurance Systems

Table 5 lists the factors influencing farmers to avoid implementing quality assurance systems. This includes the overwhelming amount of paperwork involved in the certification application process, a lack of legal support offered to the farmers during the application process, a lack of sufficient knowledge regarding quality assurance certifications, the process is time-consuming and involves a high annual cost, the lack of financial benefits, a lack of training and some feel that it is not necessary, due to their small business size. Most of these results agree with the findings of [17,18], who concluded that farmers fail to implement most agricultural technologies because they do not know them. In Ethiopia, bureaucracies are the current bottlenecks for small-scale farmers to acquire innovative ideas [13]. Ref. [19] indicated that most agricultural technologies fail because they contain high initial costs for implementation.

### 3.8. The Factors Influencing the Implementation of Rooibos Quality Assurance Systems

We used a binary logistic regression model to evaluate the relationship between independent and dependent variables. The result of the Wald chi-square test (X^2^ = 84.983, df = 8, *p*-value = 0.000) observed that predictor variables were significantly associated with the implementation of quality assurance systems (Table 6). The significant *p*-value indicated that the model was well-fitted at *p* < 0.001 level of significance. Using the Hosmer Lemeshow test, we confirmed that the model was well-fitted (*p*-value < 0.001). The log-likelihood value was 87.456, confirming an efficient model. Membership in farming societies, land tenure, farm size, off-farm income, and farmers’ education level significantly influence a farmer’s decision to implement a quality assurance system (Table 6).

## 4. Discussion

Among the eight independent variables included in the logit regression model, five appeared to significantly influence a rooibos farmer’s decision to implement a QA system. The analysis suggests that the farmers’ household’s socioeconomic factors and institutional attributes were critically important in implementing and disseminating quality assurance systems. Therefore, we shall discuss the significant independent variables and other studies that complement our findings.

### 4.1. Education Level

The farmer’s education level appeared to enhance the farmers’ decision to implement a quality assurance system. The probability of implementing a quality assurance system increased by a factor of 3.56% as the farmer’s education level increased by one year of study. This result is consistent with the findings of [13], who observed a positive association between education status and technology adoption. Moreover, as indicated by [8], a lack of education appears to be the most perilous factor limiting the dissemination of agricultural technologies such as quality assurance systems. Therefore, educating farmers about the socio-economic, environmental and health benefits of such technologies is essential. This is supported by [20,21,22,23], who noted that Africa is an emerging continent, and its growing economy is accompanied by technological development, particularly in agriculture.

### 4.2. Farm Size

There was a significant positive association between farm size and the implementation of quality assurance systems among rooibos farmers; the larger the farm, the higher the probability of quality assurance systems implementation. Farmers with more land have access to a broader range of financial services from both government and private sectors. As a result, they are more likely to have the financial means to implement quality assurance systems. The works of [24,25] noted that farmers of larger farms enjoyed better economies of scale and understood the significance of promoting and implementing new technologies. The research focused on implementing improved irrigation practices in Southern Tanzania observed a similar statistical significance between farm size and the successful implementation of farmer-led irrigation practices [26].

### 4.3. Farming Organization Membership

Being a member of a cooperative organization had a negative (significant) effect on implementing quality assurance systems among rooibos farmers. Farmers observed the advantages of the technology through the experience of early implementers within their organization. Ref. [27] concluded that having a membership to a social organization contributes to implementing agricultural technologies eventually. Few farmers will be willing to implement the technology at the onset [28]. Our finding is also supported by Rogers’ theory of implementation, where different individuals have different implementation intensities [26]. Ref. [29] observed that the time it takes for a farmer to decide whether to implement a quality assurance system or not is depended on the influence of other implementers in the social groups.

### 4.4. Land Tenure

The land tenure variable significantly affected the farmers’ decision to implement a quality assurance system. Using rented or borrowed land reduces the likelihood of implementing quality assurance systems among rooibos farmers. This finding is consistent with the results of other scholars in sub-Saharan Africa. Ref. [30] concluded that land tenure arrangements, other than ownership, are associated with diminishing security. We also reasoned that the farmers who rent the land might not control it for long enough to realize the advantages of the technological investment.

### 4.5. Off-Farm Income

We observed a statistically significant positive association between the off-farm income variable and the rooibos farmers’ implementation of quality assurance systems. This highlights the capital-intensive nature of new technologies. In essence, the off-farm income grants farmers a more comprehensive range of financial means to make the necessary decisions. This finding is in line with [31], who also noted a positive association between off-farm income and the implementation of agricultural technologies. As a result, despite existing conditions, household income is still among the most critical variables that influences a farmer’s decision-making process regarding implementing a quality assurance system [32].

## 5. Conclusions and Policy Implications

Understanding rooibos farmers’ decision-making behaviour concerning certification and the implementation of quality assurance systems is required. Several factors influence a farmer’s decision to implement a quality assurance system on their farm, including the farmers’ level of education, organization membership, farm size, land tenure, and off-farm income. Therefore, this research would help policymakers, farm managers, and the government maintains rooibos products’ quality to ensure better living standards. For instance, implementing and accepting quality assurance systems among farmers would help policymakers promote and enact efficient policy frameworks that ensure consumers are protected from harmful and low-quality products. Emphasizing a farmer’s implementation behaviour might not be enough. Building an environment that enhances the sustainable impact on various stakeholders, such as suppliers, customers, and export agencies, could likely influence the implementation of quality assurance systems.

Results from our study offer a theoretical foundation for understanding the implementation willingness of rooibos farmers. This information is vital for government departments to promote quality assurance systems and enhance decision-making.

The local governments should provide farmers with sufficient information regarding quality assurance systems, their benefits, and implementation costs. They can support the interaction between farmers through social institutions. The government should formulate targeted plans to encourage farmers to utilize quality assurance-related training, meet their needs, and improve their awareness and acceptance of quality assurance systems. This will assist the cooperatives in producing quality rooibos tea suitable for export and maintain the reputation and commercialization of the rooibos tea industry. Farming organizations or cooperative societies should inform members about the current requirements in rooibos farming too.

## Figures and Tables

**Figure 1 foods-11-03495-f001:**
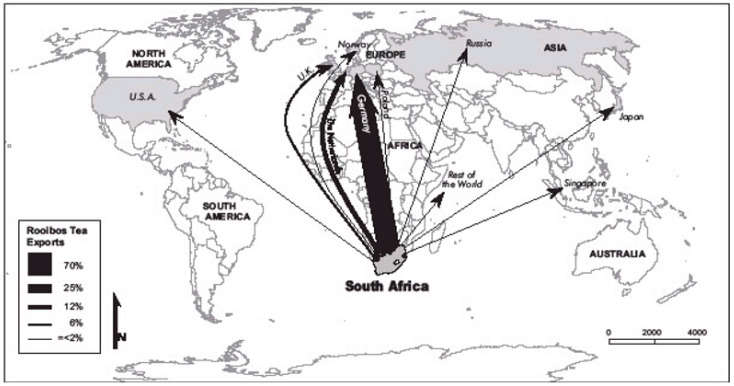
Rooibos tea export to different international markets (adapted from Olivier, J. 2008).

**Figure 2 foods-11-03495-f002:**
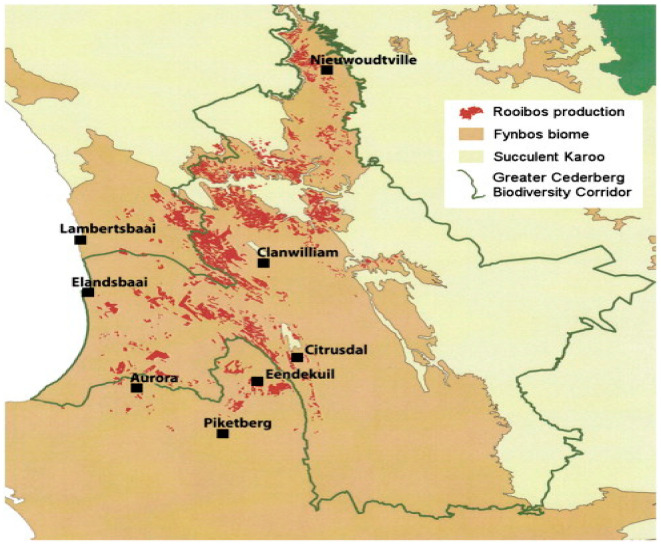
The production area of rooibos [4] (map extracted from SARC).

**Figure 3 foods-11-03495-f003:**
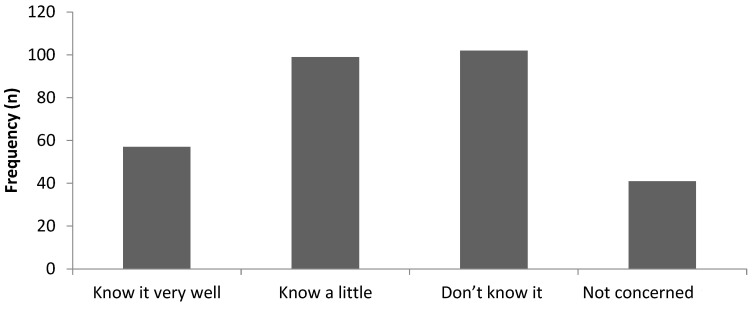
Level of rooibos farmers’ knowledge of quality assurance (QA) systems.

**Figure 4 foods-11-03495-f004:**
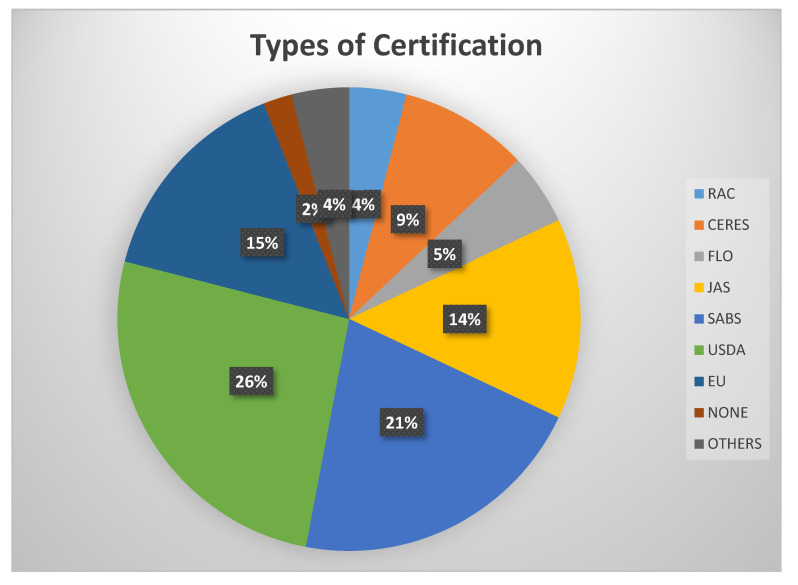
The rooibos tea farmers use different types of certifications. (RAC: Rainforest Alliance Certification; CERES: Sustainable Agricultural Networks; FLO: Flo fair trade labeling; JAS: Japanese Agricultural Standards; SABS: South Africa Bureau of Standards; USDA: United States Department of Agriculture; EU: European Union; NONE: No Certification).

**Table 1 foods-11-03495-t001:** Variable descriptions.

Variables in Equation	Description	Expected Sign +/−
QAS	Adoption of certification	+
GENDER	Gender of the household head	+/−
AGE (years)	Age of the household head	+/−
EDUC	Education level of the household head	+
HHSIZE	The number of occupants in a household	−
SOF	Size of the farm in hectares (ha)	+
TOT	Type of rooibos tea a farmer cultivates	+/−
OFFFARMINC	Household off-farm income	+
Membership	Whether a farmer participates in a farming organization/society	+

**Table 2 foods-11-03495-t002:** Participant demographic information.

Variable	Description	Percentage (%)
Education	Primary school	73.8
	High school	17.9
	College	4.0
	University	3.3
	Other formal training	1.0
Gender	Male	57.3
	Female	42.7
Farm ownership	Private	33.7
	Co-operative	21.0
	Government enterprise	9.6
	Rent	35.7
Farming organization/society	Rooibos limited	25.4
	Wupperthal cooperative	31.0
	Heiveld cooperative	22.1
	Independent farmers	21.5
Age	18–25	0.7
	26–34	2.7
	35–44	5.3
	45–54	39.7
	55–64	39.0
	Over 65	12.6

**Table 3 foods-11-03495-t003:** The different types of additional farming practices.

Farming Practice	Frequency (n)	Percentage (%)
Livestock	193	64.3
Crop (Cash/subsistence)	86	28.7
Other	21	7.0

**Table 4 foods-11-03495-t004:** Rooibos farming characteristics.

Variables	Minimum	Maximum	Mean	SD
Farm size (hectares)	2	2000	1001	-
Tea yield (tons)	700	200,000	4207.26	-
Wild tea (kg)	5	80	38.54	22.61
Cultivated tea (kg)	20	95	60.46	18.61
Wild (price/kg)	12	25	16.31	4.47
Cultivated (price/kg)	35	55	44.63	6.28

kg: kilogram; SD: Standard deviation.

**Table 5 foods-11-03495-t005:** The reasons given by rooibos farmers for failure to implement quality assurance systems.

Reason Given	Frequency (n)	Percent (%)
Lots of paperwork	100	33.30
No legal support	75	25.00
Lack of knowledge about certification	40	13.38
Time-consuming	25	8.33
High annual costs	16	5.33
Confusing terminologies	15	5.00
The small size of business	10	3.33
No financial benefits	10	3.33
Lack of training in certification	9	3.00

**Table 6 foods-11-03495-t006:** Logit model results.

Variables	B	S.E	Wald	* Sig.
Age	0.242	0.496	0.238	0.626
Gender	0.846	1.332	0.404	0.525
Hhsize	0.478	0.487	0.964	0.326
Education	0.0356	0.0564	0.498	0.027
Farm size	0.009	0.008	1.358	0.001
Membership	−2.427	0.799	9.221	0.058
Land tenure	−1.716	0.802	4.578	0.032
Off-farm Income	0.940	0.894	4.711	0.030
Constant	11.264	5.797	3.776	0.052

* Level of significance (*p* < 0.001); B: estimated logit coefficient S.E: standard error Hhsize: household size.

## Data Availability

Data is contained within the article.

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
