# Peer review of "Factors Influencing Rooibos Tea Certification and Quality Control for Smallholder Farmers in South Africa"

_foods, 2022, doi:10.3390/foods11213495_

Round 1

Reviewer 1 Report

Dear Authors,

Please find below my suggestions to improve the manuscript.

The second part of the title is not clearly understandable. Please revise it.

The sentence of lines 12 through 17 is too long. Please break it.

The introduction is a bit out of focus. The first part is very general and misses to frame the specific problem towards which this research was built. The next part goes too much into the details of the product under study. This part could be significantly trimmed, and details diverted to a “case study” section. Research aims are missing. Please clearly present the aim of the study and specify the research questions. Please introduce the methods and link them to the research questions. The identification of research gaps and the related contribution of this study are missing. Please add missing information.

Lines 79-81: In my understanding, the statement refers to tea originating from Camellia sinensis, so not to rooibos or herbal teas. The reference for Couto et al., 2021 is missing.

Please have the methods before data; this would help the reader to understand how and why the dataset for analysis was created. 

Please add a case study section

Please remove data from the results section. The results section should be structured towards analytical findings, to show how the research answers the research questions and thus achieves the aim of the study.

Figure 3: caption missing

A critical review of the overall methodological approach of the research is needed, especially to clarify the practical usefulness of research findings for advancing knowledge at a broader scale than the cases study area.

Policy implications are weak and not completely grounded on research findings. This part should be rewritten

Reviewer 2 Report

Hi dear

This article " Factors Influencing Certification and Quality Control of Rooibos Tea: A review of Smallholder Farmers in South Africa” was revised and has a novelty and consideration of the following comments.

Title: If you can rewrite and make it more interesting for readers. I propose: “Factors Influencing Rooibos Tea Certification and Quality Control: A Review of Smallholder Farmers in South Africa”.

Abstract:

·       The text is very general and brief, please express it in detail, that is, with statistics and figures.

·       The type of statistical design used in this research should be mentioned.

·       The number of statistical population should be included in the abstract.

Keywords: Please choose keywords other than the main words of the title. In this case, other researchers can find your article by searching a wide range of words through databases. I propose another keywords as the follow:

Certification, Rooibos tea, Quality assurance systems, Farming practice, Off-farm income, Education level, Influential variables, Cumulative logistic probability functions

Abbreviation:

·       Please provide “Abbreviation section consequent the Keywords

Introduction:

·       At the end of the introduction, the reason for doing this research should be described in a comprehensive paragraph

Methodology:

·       “The structured questionnaire was developed to collect important and relevant data from the respondents by way of interviews a method previously used by Rampedi and Olivier, 2008”. Please state correctly and scientifically how to refer in the entire text of the article according to the structure of the journal.

·       It seems that many factors contribute to the topic of this article (for example: Distance of cultivation place from the city, number of family members, education level of family members etc. ), why only the factors in Table 1 are enough?

Results:

·       Table 4: Please consider SD for “Farm size (hectares)” line. It seems the authors has mistaken.

Discussion:

·       Most researchers focus on its health benefits, among other advantages. Therefore, understanding farmers’ decision-making behavior concerning certification and quality assurance systems adoption among rooibos tea farmers in South Africa is essential. Please use short sentences throughout the manuscript.

Conclusions:

Conclusion is very general, try to make it more scientific, comprehensive and concise in detail, especially.

References: It is OK.

The article has many flaws in express and concept of English, it is suggested to be revised in a scientific and native way.

Reviewer 3 Report

The authors examined quality insurance of rooibos tea, an endemic product from South Africa, based on questionnaire which suggest the aptitude and attitude of the farmers toward quality assurance systems. The manuscript is interesting enough is quite clear in its objective. However, I suggest correcting some issues:   

Abstract

Lines 9-20 Please briefly add to the abstract the most relevant findings you gathered from your results.

Introduction

Line 25-26/29 Food quality and safety are not the same. Related, but not the same. Please rephrase for clarity. I do not quite see the relationship between the results of your study and food safety.

Line 52 It seems you missed a reference here, it is just labeled as “(REF)”.

Line 52-54 This idea rounds up neatly relationship between quality and safety.

Line 58 The “L” in Aspalathus linearis should be in lower case, also the name of the researcher who first described the plant, (Burm. f.) R. Dahlgren, should be included.

Line 64-65 It would be interesting to include how much these exports represent in economic terms for the country. How much products is exported or produced?

Line 71 How large is the South African tea market and how does it compare with other major tea producing countries.

Line 72 How much does 10% represent in mass?

Research methodology

Line 97 Most reader may not be familiarized with this area of South Africa I suggest including the coordinates/latitude and longitude.

Line 109 What do the authors mean with “tea rejection”? Consumers rejecting or returning the product for some reason? Please elaborate

Line 113 please abstain to use “etc”. Please rephrase

Line 171 Please elaborate as to what “Rand” means. Your readership could not be familiarized with this term.

Line 181 I do not think television would be a good reference where people could learn about food safety access to world wide web would be more relatable in this day on age.

Line 242 level of significance.

I suggest to add a section dedicated that include which statistical tests were used to assess each variable. This will help the reader in the long run understand what was done.  

Discussion

Line 263 “Judgments”? Which judgment? From whom? Please clarify

Line 277 Please explain again how cooperative society work has positive and negative effects? These are not clear. Also, what exactly is significant here? What alfa was used?             

Figure 3 could be cleaner. A white background could help. I also suggest putting the color key in the figure legend. Please change the “note” for the correct figure legend for this figure which is missing.

Line 310 health benefits from whom? I guess, the tea? An interesting issue arise here, a poor-quality assurance program for the product may undermine any health benefits the product itself can generate.

Reviewer 4 Report

This study  aims to determine factors that influence adoption of quality assurance system among rooibos tea farmers in South Africa. The authors evaluate the challenges regarding shifting requirements  current rooibos tea farmers are facing significant and demand for quality and safety of rooibos tea  by-products.

This research is important and can bring valuable information with practical application. 

The presented research is well-planned, and the manuscript is generally well organized. Overall, the work is well written and organized, although there are some typing errors to correct during the revision of the work and some methodological (statistical in special) data are missing. 

It must be improved English language. Moreover, some sentences are  unclear.

The full name of the species is Aspalathus linearis (N.L.Burm.) R.Dahlgr.,  Fabaceae family - it must be mentioned in full at the first introduction in the text.

The Introduction provides enough data on the stage knowledge of these issues. but in the introduction, the research design is not presented, with a clear presentation of the steps and methods used for answering the research question. Novelty and originality should clearly be stated at the final of chapter, in order to state what your study brings in novelty. All these should be highlighted in order to increase the value of the obtained results.

More data related to the statistical methods used are needed.

Table 6 (Logit model results) has incomplete data, the abbreviations must be explained in the Notes, the level of significant* are not specified. Also, additional explanations can be added to the other tables and figures in the Notes.

In chapter Conclusion and policy implications the research conclusions and solutions should be formulated more clearly and concretely.
